# Health Needs Assessment: Chronic Kidney Disease Secondary to Type 2 Diabetes Mellitus in a Population without Social Security, Mexico 2016–2032

**DOI:** 10.3390/ijerph19159010

**Published:** 2022-07-25

**Authors:** Silvia Martínez-Valverde, Rodrigo Zepeda-Tello, Angélica Castro-Ríos, Filiberto Toledano-Toledano, Hortensia Reyes-Morales, Adrián Rodríguez-Matías, Juan Luis Gerardo Durán-Arenas

**Affiliations:** 1Centro de Estudios Económicos y Sociales en Salud, Hospital Infantil de México Federico Gómez, Instituto Nacional de Salud, Mexico City 06720, Mexico; smartinez@himfg.edu.mx; 2Dirección de Prestaciones Económicas y Sociales, Instituto Mexicano del Seguro Social, Mexico City 06600, Mexico; rodrigo.zepeda@imss.gob.mx; 3Unidad de Investigación en Epidemiología Clínica, Hospital de Pediatría, Centro Médico Nacional SXXI, Instituto Mexicano del Seguro Social, Mexico City 06720, Mexico; 4Unidad de Investigación en Medicina Basada en Evidencias, Hospital Infantil de México Federico Gómez, Instituto Nacional de Salud, Mexico City 06720, Mexico; filiberto.toledano.phd@gmail.com; 5Centro de Investigación en Sistemas de Salud, Instituto Nacional de Salud Pública, Cuernavaca 62100, Mexico; hortensia.reyes@insp.mx; 6Servicio de Nefrología, Hospital Angeles Metropolitano, Mexico City 06760, Mexico; adrianrodmati@hotmail.com; 7Departamento de Salud Pública, Facultad de Medicina, Universidad Autónoma de México, Mexico City 04510, Mexico; lduranarenas@gmail.com

**Keywords:** chronic kidney disease, health needs, Markov model, diabetes mellitus, Mexico, assessment

## Abstract

Health needs assessment is a relevant tracer of planning process of healthcare programs. The objective is to assess the health needs of chronic kidney disease (CKD) secondary to type 2 diabetes mellitus (T2 DM) in a population without social security in Mexico. The study design was a statistical simulation model based on data at the national level of Mexico. A stochastic Markov model was used to simulate the progression from diabetes to CKD. The time horizon was 16 years. The results indicate that in 2022, kidney damage progression and affectation in the diabetic patient cohort will be 34.15% based on the time since T2 DM diagnosis. At the end of the 16-year period, assuming that the model of care remains unchanged, early renal involvement will affect slightly more than twice as many patients (118%) and cases with macroalbuminuria will triple (228%). The need for renal replacement therapy will more than double (169%). Meanwhile, deaths associated with cardiovascular risk will more than triple (284%). We concluded that the clinical manifestations of patients with CKD secondary to T2 DM without social security constitute a double challenge. The first refers to the fact that the greatest health need is early care of CKD, and the second is the urgent need to address cardiovascular risk in order to reduce deaths in the population at risk.

## 1. Introduction

Health needs assessment consists of identifying the health or illness conditions in a population that is likely to benefit from healthcare, including disease prevention, diagnosis, treatment, rehabilitation and general medical care [1,2,3]. As per this definition, needs are the basis for describing a “potential demand for care,” a central point for the planning process of healthcare programs [4].

In the context of health needs assessment, chronic kidney disease (CKD) is a relevant tracer because of its side effects and its high prevalence worldwide. Type 2 diabetes mellitus (T2 DM) is the most common contributor to CKD and is the first cause for entry into renal replacement treatment programs (e.g., dialysis and transplantation). In high-risk populations, more than 40% of individuals with T2 DM develop CKD [5,6]. More than 5% of patients newly diagnosed with T2 DM showed evidence of renal involvement and are at a risk of developing nephropathy 10 years after diagnosis [7,8].

Likewise, CKD represents an increased risk of death due to cardiovascular disease (CVD) [7,8,9]. CVD is the primary cause of premature mortality among patients with T2 DM who have CKD (50.3%). The United States Renal Data System (USRDS) 2014 Annual Report reinforces this fact and states that the prevalence of any CVD is double in patients with CKD than in the general population (69.8% vs. 34.8%) [10,11].

Therefore, the timely diagnosis of CKD at an early stage could prevent or delay relevant health consequences and their economic burden [12,13]. Given that end-stage renal replacement treatment requires specialized human and technological facilities, it represents an economic burden for the health system and their patients.

The concern regarding CKD secondary to T2 DM in the healthcare system has been focused on its complications, which primarily appear as anemia, malnutrition, dyslipidemia, metabolic acidosis, hypertension, osteodystrophy and events associated with the cardiovascular system [14,15]. These complications decrease the patient’s quality of life and increase pressure on the demand for service, thus affecting the economic increment in medical care.

International estimates of CKD range from 10% to 16% of the adult population. In a 12-country report involving over 75,000 individuals, the prevalence of CKD was 14% among the general population [16]. Studies in the United States have reported a prevalence of early stage CKD in 14.5% of the adults, which implies that 30 million American adults have CKD. In 2015, 124,111 new cases of end-stage renal disease were reported, which represents an incidence of 527 per million people [11].

In Mexico, the prevalence of CKD is underestimated because studies are only focused on end-stage renal disease. There is no registry for renal patients to quantify the magnitude of CKD secondary to T2 DM [17]. However, it could be assumed that the prevalence of CKD is increasing, given the fact that based on data from the National Health Surveys, the prevalence of T2 DM in Mexico reached 13.5% (9.5% diagnosed and 4.1% undiagnosed) on average in the adult population in 2016 [18] and it increased to 10.3% in diagnosed people in 2018 [19]. Therefore, it is safe to assume that T2 DM has strongly influenced the epidemiology of CKD in recent years.

In Mexico, CKD is usually identified late in patients with diabetes during advanced stages of the disease at second-level general hospitals [20]. There are no strategies for the effective management of patients with T2 DM; therefore, the quality of healthcare for this disease in Mexico is compromised. In 2016, 81.3% of patients of T2 DM without social security had uncontrolled glycemic levels despite receiving pharmacological treatment [21]. So, these patients could present greater complications of T2 DM and a more accelerated progression to end-stage renal.

End-stage renal disease represents an inequality in access to health services because a small proportion of patients are provided with appropriate renal replacement treatment [22]. People who are not affiliated to social security health services have been the most unprotected because this population does not have regular access to renal replacement treatment due to the lack of coverage in the package of benefits as well as the high financial costs for the families.

In Mexico, the provision of end-stage renal treatment includes access to healthcare services for those who benefit from the social security system, such as workers and their families who serve in the private industrial formal sector and all public government employees (by law, these users have access to basic benefits that all employers must provide) [23].

Although a significant proportion of the Mexican population is covered by the public healthcare system “Instituto de Salud para el Bienestar” (INSABI), (the financial mechanism for the open population), previously known as the “Social Protection System in Health” (SSPS), renal replacement therapy is not included in the benefits of the public institutions network under the Ministry of Health management [24]. This group includes all of those without a fixed or formal job temporarily or permanently.

In this healthcare context, detecting the health requirements of CKD secondary to T2 DM disease at early and intermediate stages is important because its consequences could be prevented, thereby reducing the progress of the disease. One alternative for the assessment of health needs of CKD secondary to T2 DM in the Mexican population without social security is to apply statistical techniques based on the T2 DM prevalence in Mexico to estimate the need. In this article, we present the results of such an exercise.

## 2. Materials and Methods

The study design was a statistical simulation model based on data at the national level of Mexico. A stochastic Markov model was used to simulate the progression from diabetes to CKD. Hypothetical cohorts [25] of patients with T2 DM without social security in Mexico were entered in the model. The time horizon was 16 years. The prevalence of T2 DM based on age group and years with T2 DM since diagnosis was obtained from ENSANUT MC 2016 [18]. Health needs assessment was estimated as the number of cases for each stage of CKD secondary to T2 DM.

### 2.1. Simulated Cohorts

To define the prevalence cohorts, the percentages of T2 DM prevalence in 2016 were extrapolated to the population of Mexico [26] based on age group (20–29, 30–39, 40–49, 50–59, 60–69 and 70–79 years of age). The target population was calculated as the percentage of people who report not having social security and receive health services in the public institutions network under the Ministry of Health, such as the SSA. The estimates were arrived at as follows:
(1)Health needs assessment = T2 DM cases in 2016 that progressed to CKD.(2)T2 DM cases for year i = Prevalence of T2 DM in 2016 + 
$\underset{k\leq i}{\sum }\mathrm{Incidence}\;\mathrm{of}\;T2\;\mathrm{DM}\;\mathrm{for}\;\mathrm{year}\;i.$
(3)Population at risk (non-diabetic adults) for year i was assessed with the total Mexican population for year i—Prevalence of T2 DM in 2016.(4)Incidence of T2 DM for year i = Incidence rates of T2 DM for year i x population at risk for year i.(5)Percentage of patients with DM with glycemic control (HbA1c < 7) in primary healthcare centers in 2016 = Diabetes patients with glycemic control HbA1c < 7%/patients with diabetes in treatment.

### 2.2. Information Sources

The Mexican population in the age group of 20–79 years for 2016–2032 was obtained from the population projections of the National Population Council (Consejo Nacional de Población; CONAPO) [27]. The incidence rates of Diabetes were from the literature Meza, R et al., 2015 [28].

People who reported as not having social security were reviewed in the National Survey on Employment and Social Security (Encuesta Nacional de Empleo y Seguridad Social- ENESS) 2017. It is a representative sample of the national population [29].

Diabetes prevalence was taken from the literature Basto-Abreu A et al., 2019 [30]. T2 diabetes prevalence and years since diagnosis (Table 1) were from 2016, ENSANUT MC Database [18]. The percentage of patients with DM (HbA1c < 7) in primary healthcare centers in 2016 was from the Observatorio Mexicano de Enfermedades Crónicas No Transmisibles (OMENT) [21].

### 2.3. CKD Progression Model

Definition of the illness: CKD was defined as abnormalities in the renal structure or function present for >3 months and was based on an albuminuria marker [31], albumin excretion rate of ≥30 mg/24 h and albumin/creatinine ratio of ≥30 g/g (≥3 g/mol). The parameter used to measure renal involvement and cardiovascular risk in patients with T2 DM was albuminuria [32].

A model was used to determine the progression via a discrete time Markov chain [33,34]. The progression was replicated in prevalent cohorts and incident cases of T2 DM for each year. The initial condition of CKD secondary to T2 DM was established based on the average years with diagnosis and the percentage of patients with T2 DM control (18.7%) of glycemic (HbA1c < 7) and uncontrolled T2 DM in the primary healthcare centers [21].

Model structure of CKD secondary to T2 DM (Figure 1. Health States) [35]:

(A) Normoalbuminuria < 30 mg/g; (B) Microalbuminuria 30–300 mg/24; (C) Macroalbuminuria ≥ 300 mg/g; (D) End-stage renal disease (renal replacement); (E) Cardiovascular death.

We randomly assigned the time since first diagnosis following the times reported by ENSANUT and ran the model since that diagnosis. We modeled the probability via kernel density estimation. The transition probability distributions were obtained from the United Kingdom Prospective Diabetes Study (see Table 2) [35].

To assess the model, we ran simulations of individuals and contrasted the expected number of patients per state vs. analytical results for the expected value. The number of simulations was estimated as the minimum sample size to obtain an error of <10,000 and rounded off. Quantiles, median and variance were obtained from the simulations. The model was programmed using R 4.0.3 statistical package, R Core Team (2020), R Foundation for Statistical Computing, Vienna, Austria.

## 3. Results

Table 3 identifies different degrees of renal progression in cohorts of patients with t2 DM without social security in Mexico. During the time frame of 2016–2032, the progression of kidney damage could affect around 1.4 to 4.4 million people, considering the parameters of the model constant.

.

In 2022, kidney failure progress in the diabetic patient cohort will be 34.2%. According to case distribution by stages (see Figure 2), 12% (850,000 patients) are characterized as being at an early stage, with incipient damage but with an increased cardiovascular risk due to the presence of microalbuminuria; 3% (214,000 patients) transit through macroalbuminuria, which implies a deterioration of renal function; 0.65% (46,000 patients) require kidney substitution therapy; and 18.5% (1.3 million patients) died from some cause associated with cardiovascular damage, which is typical of T2 DM and CKD.

In 2024, the kidney damage progression increases to 36.4% of the cohort, affecting 2.779 million patients with T2 DM (see Table 3); however, in the evolution of the disease the dominant outcome refers to the death of 20% of the cohort.

At the end of the 16-year period, in 2032, kidney damage progression will be 43% with 4.4 million of patients. According to case distribution, microalbuminuria cases will increase from 10.6% to 12.40%, macroalbuminuria cases will range from 2.2% to 3.8% and patients requiring kidney substitution therapy will account for 0.53–0.76% of the cases. The outcome with the highest growth is deaths with 2.679 million, representing 26.2% of the total affected.

Figure 3 shows the evolution of CKD secondary to T2 DM over time. At the end of the 16-year period, assuming that the model of care remains unchanged, early renal involvement will affect slightly more than twice as many patients (118%) and cases with macroalbuminuria will be tripled (228%). The need for renal replacement therapy will more than double (169%). Meanwhile, deaths associated with cardiovascular risk will more than triple (284%). It should be noted that these estimates include the 22% growth of the Mexican population predicted by CONAPO.

In the Appendix A are shown the probabilities estimated from evolution of CKD secondary to T2 DM for each cohort by age group, at each stage of the disease (normoalbu-minuria, microalbuminuria, macroalbuminuria and terminal CKD) and for min-max values, variance and quartile of variation.

## 4. Discussion

The first relevant result of this study is the identification of a preventable excess mortality. The mortality rate has almost tripled in less than two decades, thus evidencing the need to give attention to cardiovascular risk in patients with CKD secondary to T2 DM.

The second aspect refers to the urgent need to focus on the early stages of CKD. The second largest concentration of the population with diabetes has a preventable health condition (microalbuminuria). This indicates a segment of opportunity to delay progression to late stages so as to achieve a longer time with quality of life and lower costs of care in the health system.

The third point identifies that the need for renal replacement therapy has more than doubled, reaching 77,000 cases. In the future, it represents a significant potential demand for dialysis and hemodialysis services that are currently under increased pressure.

The estimates of this work, in comparison with studies conducted in Mexico and in the global context, are consistent in agreeing that the burden of the disease is in the early stages of CKD.

In Mexico, the Kidney Early Evaluation Program (KEEP) reported a CKD incidence of 14% in 2013, with stages 1 and 2 being more frequent [36]. Another empirical study reported a prevalence of abnormal albuminuria of 13% in two cities in Mexico in 2013 [37]. Worldwide, a systematic review of observational studies reported annual incidences of CKD ranging from 3.8% to 12.7% for microalbuminuria, 1 to 4.8% for macroalbuminuria and terminal CKD from 0.04% to 1.8% [38].

In Mexico, according to the National Institute of Statistics and Geography (INEGI) on the leading causes of death in 2020, heart disease was the leading cause of death, with 218,704 deaths (20.1%), followed by COVID-19 with 200,256 (18.4%). Deaths associated with diabetes mellitus were 151,019 (13.9%), while deaths from terminal CKD were 6618 cases [39]. These numbers of early and avoidable deaths suggest that prevention and care services in Mexico are not sufficient or are not focused on those at the highest risk.

In the medium term, improving CKD care is expected to be a feasible goal through targeting programs that modify the trajectory of cardiovascular disease in patients with CKD secondary to diabetes and its health consequences. Without addressing CKD and investing in modifying trajectories, early and preventable deaths will continue to occur due to the increased cardiovascular risk that evolves throughout the natural history of CKD and diabetes [40,41], as well as the presence of risk factors that accelerate atherosclerosis and progression of CKD that prompts premature mortality before evolving to late-stage CKD [10,42,43].

Patients with CKD have comorbidities that increase the death risk. Some of the general comorbidities are high blood pressure, dyslipidemia and atherosclerotic disease. In addition, there are other specific comorbidities in CKD such as left ventricular hypertrophy, low serum albumin levels, elevated serum phosphate and hemoglobin levels below the international goals for chronic kidney disease [44]. One of our limitations in this study is the lack of clinical data on these specific variables; however, we are considering a second study in which an effort will be made to measure them.

One of the most important actions to address the problem is to identify the latent risks of CKD in the population with uncontrolled diabetes without social security.

For this purpose, it is necessary to plan preventive actions that allow timely identification of renal function deterioration through intentional random screening of urine albumin and/or estimation of glomerular filtration rate (GFR), as well as the modification of risk factors and establishment of pharmacological and non-pharmacological lines of management that contain the growth of CKD. All preventive actions should be accompanied by nutritional management to achieve better glycemic control and decrease cardiovascular risk [45].

In agreement with the definition of KDIGO International guidelines, and after the adaptation to the clinical practice in the health system of Mexico, it is suggested that albuminuria evaluations in patients with diabetes are needed to estimate cardiovascular risk and its complications; on the other hand, it is suggested that the GFR could be used as a complementary measure to evaluate the existence of hyperfiltration in the diabetic patients.

According to Molitch et al. [46], albuminuria reflects glomerulopathy along with measures of glomerular filtration. People with diabetes may only develop albuminuria, only decreased glomerular filtration or both. Independent of albuminuria and diabetes, measures of glomerular filtration predict CKD. Both measures independently increase the risk of mortality [46].

The lack of glycemic control in diabetic patients has an impact on the accelerated progression of CKD, so it is necessary to emphasize the need to invest in education programs for diabetic patients in order to promote their participation in the control of the disease, such as the self-monitoring of glucose, with patients being taught how to identify adequate control figures. In the same way, it is necessary to focus on programs that address the patient’s beliefs in order to improve self-management and self-motivation of their disease and risks, as well as to increase the commitment to pharmacological adherence, which is beneficial in the long term to avoid and delay microvascular complications [47].

For the preventive actions, it could be important to consider the economic aspects of these interventions, given that in Mexico the attention has been concentrated on pharmacological therapy and not preventive measures.

With regard to access to substitution therapy, this is of great importance as it positively affects the patient’s quality of life, with a consequent decrease in mortality. In a study of 790 patients without social security who received renal replacement therapy at the General Hospital of Mexico, the evidence showed an overall survival of 45.8% at 3 years. By type of therapy modality, survival of patients who received renal transplantation was 100%, while for patients who entered a chronic hemodialysis program it was 61.9%, for those who received continuous ambulatory peritoneal dialysis it was 72.5% and for patients with intermittent hemodialysis it was 8.2% [48].

At present, renal replacement therapy in Mexico is not included in the package of benefits of the health system for patients who lack social security and who are cared for at the Health Institute for Welfare (INSABI). As mentioned before, INSABI is the organization created by the current Mexican government to replace the People’s Health Insurance (Seguro Popular), which had affiliated 50% of the country’s population until 2018; however, the program has failed to provide the services that the population demands, the most important being renal replacement therapy [49].

Another relevant aspect in mortality is the inequalities in the healthcare of patients with renal replacement therapy with and without social security as the latter have a higher mortality rate (56% vs. 38.2%) [48].

An alternative for patients in need of renal treatment would be to offer access to care through temporary inclusion in social security services that not only include the coverage of the substitutive treatment but also care of the underlying disease complex. This implies the allocation of a budget, changes in the social security law and the universalization of healthcare for this group of patients in Mexico.

Another modality to decrease the differences in mortality is to finance the production of renal replacement therapy and increase the coverage of public systems. “Disparities in health care exacerbate the negative effects of biological predisposition and the increased burden of CKD in disadvantaged populations, due to low socioeconomic status, and poor access to health care” [50].

The most significant limitation is the lack of available data on the progression from diabetes to CKD in the Mexican population. Because of this, the model assumes that the population diagnosed with T2 DM is under treatment and has followed a similar trajectory as the individuals in the UKPDS study. This assumption implies that the estimations on CKD secondary to T2 DM may be underestimated with respect to the real need in the Mexican health system. Hence, we used the highest transition rates from the UKPDS study to model uncontrolled diabetes in order to reflect the health impact and quality of care in Mexico.

Another limitation of the model concerns conventional diabetic nephropathy through the transition from normal levels of albuminuria to microalbuminuria, macroalbuminuria, terminal CKD and death. However, over the past few years, it has been described in the literature that certain diabetic patients did not present the same evolution. For example, some of them have significant initial deterioration of glomerular filtration rate whereas, in others, microalbuminuria is reduced spontaneously, or else, patients with normoalbuminuria have progressive renal insufficiency, referred to as normoalbuminuric diabetic kidney disease (NADKD), which was not included in the model [51,52].

Another issue that was left out was the analysis by gender. The scientific literature reports that sex is a variable that has a bearing on the progression of CKD, for example, the male sex is associated with worse CKD progression. Unfortunately, the information of CKD by gender is not available, so we modeled both groups using the same progression rates [53].

Future research should assess the effect of other chronic diseases in population groups with additional risks and estimate the effect of early and timely care on the demand of renal substitutive therapy.

## 5. Conclusions

The clinical manifestations of patients with CKD secondary to T2 DM without social security constitute a double challenge for the Mexican healthcare system. The first refers to the fact that the greatest health need is early care of CKD and the second is the urgent need to address cardiovascular risk in order to reduce deaths in the population at risk.

## Figures and Tables

**Figure 1 ijerph-19-09010-f001:**
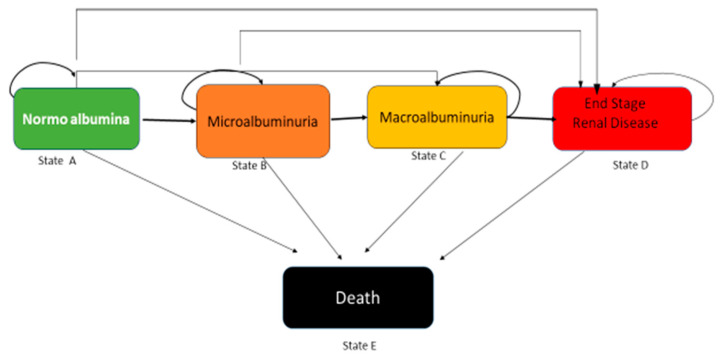
The direction of the arrows symbolizes the proportion of patients progressing toward the different stages. The returning arrows symbolize the proportion of patients who remained in each stage.

**Figure 2 ijerph-19-09010-f002:**
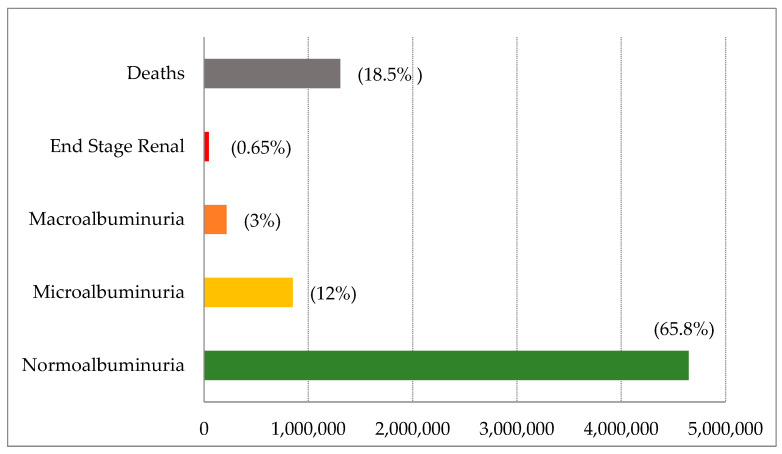
Distributions of CKD stages secondary to T2 DM, 2022.

**Figure 3 ijerph-19-09010-f003:**
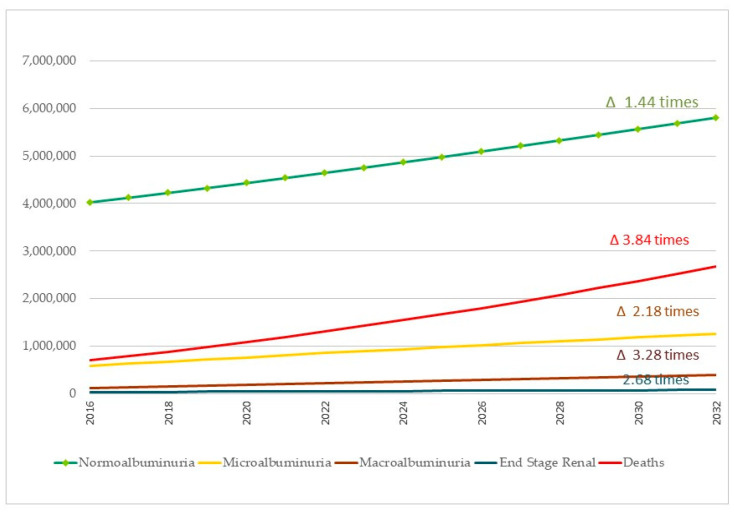
Health needs of CKD secondary to T2 DM in the Mexican population without social security.

**Table 1 ijerph-19-09010-t001:** Type 2 diabetes prevalence and years since diagnosis.

Total Prevalence of Diabetes in Mexico in 2016/*1		Years Since Diagnosis in Population Without Social Security/*2		
Age Group	(% Diagnosed and Undiagnosed)	Sum of Weights	Median [Years with Diabetes]	Interquartil Range [25%–75%]
20–29	3.3	65,962	2	[1–2 years]
30–39	3.2	127,191	5	[4–7 years]
40–49	13.8	562,243	5	[3–9 years]
50–59	26.9	946,925	9	[2–16 years]
60–69	36.5	1,185,025	10	[5–13 years]
70–79	25.5	263,384	15	[10–22 years]

/*1 Source: Basto Abreu et al. [30]; /*2 Source: Authors’ estimations based on ENSANUT MC 2016, INSP México [18].

**Table 2 ijerph-19-09010-t002:** Transition probabilities.

	Health Stages (Transition)	95% Confidence Intervals (Min and Max)
**Annual transition** **rates of** **nephropathy**	Normoalbuminuria to microalbuminuria	[1.9% to 2.2%]
	Microalbuminuria to macroalbuminuria	[2.5% to 3.2%]
	Macroalbuminuria to end-stage renal disease (renal replacement)	[1.5% to 3.0%]
	No nephropathy to death	[1.3% to 1.5%]
	Microalbuminuria to death	[2.6% to 3.4%]
	Macroalbuminuria to death	[3.6% to 5.7%]
	End-stage renal disease to death	[14% to 24.4%]

Source: Adler AI et al. [35]. “Adapted with permission from Ref. [35]. Copyright 2003, copyright Adler, A. I.”.

**Table 3 ijerph-19-09010-t003:** Health needs assessment of CKD secondary to T2 DM in patients without social security in Mexico (20–79 years of age).

Health Needs	^a/^2016	^b/^2017	^b/^2018	^b/^2019	^b/^2020	^b/^2021	^b/^2022	^b/^2023	^b/^2024
Diabetes population without social security	5,449,204	5,703,343	5,963,455	6,229,519	6,501,512	6,779,409	7,063,191	7,352,844	7,648,353
**Cases of chronic kidney disease secondary to T2 DM**									
Normoalbuminuria	4,026,612	4,124,684	4,225,015	4,327,494	4,432,015	4,538,472	4,646,773	4,756,831	4,868,561
Microalbuminuria	578,720	626,333	672,847	718,388	763,074	807,011	850,296	893,015	935,250
Macroalbuminuria	118,084	133,416	149,127	165,159	181,460	197,987	214,704	231,581	248,592
End-stage renal	28,750	31,562	34,415	37,307	40,236	43,199	46,194	49,216	52,264
Deaths associated with CV risk.	697,039	787,347	882,051	981,171	1,084,727	1,192,739	1,305,225	1,422,201	1,543,686
Distribution									
Normoalbuminuria	74%	72.3%	70.8%	69.5%	68.2%	66.9%	65.8%	64.7%	63.7%
Microalbuminuria	10.6%	11.0%	11.3%	11.5%	11.7%	11.9%	12.0%	12.1%	12.2%
Macroalbuminuria	2.2%	2.3%	2.5%	2.7%	2.8%	2.9%	3.0%	3.1%	3.3%
End-stage renal	0.53%	0.55%	0.58%	0.60%	0.62%	0.64%	0.65%	0.67%	0.68%
Deaths	12.8%	13.8%	14.8%	15.8%	16.7%	17.6%	18.5%	19.3%	20.2%
**Progression to CKD (cases)**	** ^b/^ ** **2025**	** ^b/^ ** **2026**	** ^b/^ ** **2027**	** ^b/^ ** **2028**	** ^b/^ ** **2029**	** ^b/^ ** **2030**	** ^b/^ ** **2031**	** ^b/^ ** **2032**	
Cohort of DM T2	7,949,693	8,256,841	8,569,754	8,888,350	9,212,529	9,542,175	9,877,143	10,217,299	
Normoalbuminuria	4,981,876	5,096,690	5,212,900	5,330,372	5,448,955	5,568,490	5,688,796	5,809,707	
Microalbuminuria	977,072	1,018,546	1,059,731	1,100,681	1,141,441	1,182,050	1,222,543	1,262,945	
Macroalbuminuria	265,716	282,938	300,243	317,622	335,065	352,566	370,121	387,724	
End-stage renal disease	55,335	58,425	61,533	64,657	67,795	70,945	74,108	77,280	
Deaths associated with CV risk	1,669,695	1,800,243	1,935,346	2,075,018	2,219,273	2,368,122	2,521,576	2,679,643	
Distribution									
Normoalbuminuria	62.7%	61.7%	60.8%	60.0%	59.1%	58.4%	57.6%	56.9%	
Microalbuminuria	12.29%	12.34%	12.37%	12.38%	12.39%	12.39%	12.38%	12.40%	
Macroalbuminuria	3.34%	3.43%	3.50%	3.57%	3.64%	3.69%	3.75%	3.79%	
End-stage renal disease	0.70%	0.71%	0.72%	0.73%	0.74%	0.74%	0.75%	0.76%	
Deaths	21.00%	21.80%	22.58%	23.35%	24.09%	24.82%	25.53%	26.23%	

Source: Authors’ estimates. ^a/^ Health Needs Assessment = T2 DM cases in 2016 that progressed to CKD; ^b/^ T2 DM cases (2017…2032) = Prevalence of T2 DM_(2016)_ + 
$\underset{k\leq i}{\sum }\mathrm{Incidence}\;\mathrm{of}\;T2\;\mathrm{DM}\;\mathrm{for}\;\mathrm{year}\;i$
.

## Data Availability

The generated data is available in the Appendix A Section, the corresponding citation is appreciated.

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
