# Peer review of "Health Needs Assessment: Chronic Kidney Disease Secondary to Type 2 Diabetes Mellitus in a Population without Social Security, Mexico 2016–2032"

_ijerph, 2022, doi:10.3390/ijerph19159010_

Round 1

Reviewer 1 Report

Authors

Health needs assessment is considered important as a tracer in the planning process of health care programs. Therefore, this paper aimed to assess health needs for chronic kidney disease (CKD) secondary to type 2 diabetes mellitus (T2 DM) in an area of Mexico without social security. By assessing health needs for CKD, this paper concludes that early treatment of CKD is most needed and that there is an urgent need to address cardiovascular risk in order to reduce deaths among those at risk. Specific comments from the reviewers are listed below.

Comments:

1. This health needs assessment is only for diabetes, but what are the authors' thoughts on whether diseases other than diabetes (glomerulonephritis and glomerulosclerosis), which are the primary diseases of CKD, do not need to be assessed?

2. The CKD progression model seems to assess albumin, albumin/creatinine, etc. Is glomerular filtration rate not necessary to assess CKD, or what are the authors' thoughts?

3. Why is this health needs assessment not compared between men and women?

Author Response

Comments:

  1. This health needs assessment is only for diabetes, but what are the authors' thoughts on whether diseases other than diabetes (glomerulonephritis and glomerulosclerosis), which are the primary diseases of CKD, do not need to be assessed?

Answer: We agree with this comment, however, diabetes is the number two cause of dead in Mexico. So, we decided to focus on diabetes because of its relative importance in Mexico, in terms on health and economics.

  1. The CKD progression model seems to assess albumin, albumin/creatinine, etc. Is glomerular filtration rate not necessary to assess CKD, or what are the authors' thoughts?

Answer: Yes, we think that Glomerular filtration rate is necessary to assess. In discussion section, page 3, 3rd paragraph we discuss that it is necessary to assess the glomerular filtration rate in the patients. “ it is necessary to plan preventive actions that allow timely identification of renal function deterioration through intentional random screening of urine albumin and/or estimation of glomerular filtration rate (GFR), as well as modification of risk factors and establishment of pharmacological and non-pharmacological lines of management that contain the growth of CKD.”

In addition, we include in the discussion section that it is necessary to evaluate the measurement of albuminuria in urine as well as the glomerular filtration rate because they are complementary measures.

  1. Why is this health needs assessment not compared between men and women?

Answer: We agree with the reviewer and we added a paragraph in discussion section. “The scientific literature reports that sex is a variable that has a bearing on the progression of CKD, male sex is associated with worse CKD progression. Unfortunately, the information about prevalence of CKD by sex and age is not available for Mexico, so we modelled using the same progression rates. “

Moreover, we don’t have information of the estrogenic levels in feminine population: Valdivielso JM, Jacobs-Cachá C, Soler MJ. Sex hormones and their influence on chronic kidney disease. Curr Opin Nephrol Hypertens. 2019 Jan;28(1):1-9. doi: 10.1097/MNH.0000000000000463. PMID: 30320621.

Reviewer 2 Report

The manuscript of Martinez-Valverde describes a model for the progression of chronic kidney disease (CKD) in a Type 2 Diabetes mellitus (T2DM) population in Mexico where social security and health care can be sparse. This is very important work that should be alerting all levels of government of the critical need of care for these patients that will inevitably succumb to cardiovascular disease (CVD) and complete kidney failure. The authors have used computer modelling to predict the numbers of cases and deaths resulting from untreated T2DM and CKD. The data modelling and statistics are reasonable and satisfactory. The manuscript is well-written, well-researched, using proper citations.  Conclusions are well-supported by the data. I have only minor comments.

1) The model takes into account T2DM causing CKD and then either kidney failure or CVD death. This is fair and accounts for a significant amount of cases.  However, even if not included in the model, the authors should introduce and discuss other complications of T2DM (including metabolic syndrome, hyperlipidemia, atherosclerosis, non-alcoholic fatty liver disease, high blood pressure) that may contribute to deaths. It is possible that some of these other complications or co-incident conditions to T2DM may also contribute to CKD. 

2) While I can appreciate the purpose of this manuscript, to alert the government of an impending health crisis, among other things, it seems that encouraging treatment for T2DM patients is not part of the model. Simple "treatments" such as glibenclamide (insulin sensitizer) costs about $0.03 per tablet (or less) and simple instruction on diet alterations are a very effective way to eliminate or delay T2DM complications. It would be very informative if the authors could include/discuss what a very simple and cost-effective plan could do to reduce morbidity and mortality. This is something that the Mexican government and many governments around the world would appreciate (ie. if we spend 20 million dollars on education programs, we reduce or delay this many dialysis visits (and thereby save money!), and prevent this many deaths).  

Author Response

Reviewer 2.

  • The model takes into account T2DM causing CKD and then either kidney failure or CVD death. This is fair and accounts for a significant amount of cases. However, even if not included in the model, the authors should introduce and discuss other complications of T2DM (including metabolic syndrome, hyperlipidemia, atherosclerosis, non-alcoholic fatty liver disease, high blood pressure) that may contribute to deaths. It is possible that some of these other complications or co-incident conditions to T2DM may also contribute to CKD.

Answer: Thank you, we agree with the comment and we added this paragraph in the limitations section:

“Patients with CKD have comorbidities that increase the death risk. Among the general comorbidities are included: High blood pressure, dyslipidemia, and athero-sclerotic disease. In addition, there are other specific comorbidities in CKD such as left ventricular hypertrophy, low serum albumin levels, elevated serum phosphate, and hemoglobin levels below the international goals for chronic kidney disease. [44] One of our limitations in this study is the lack of clinical data on these specific variables, however we are considering a second study in which an effort will be made to measure them.”

  • While I can appreciate the purpose of this manuscript, to alert the government of an impending health crisis, among other things, it seems that encouraging treatment for T2DM patients is not part of the model. Simple "treatments" such as glibenclamide (insulin sensitizer) costs about $0.03 per tablet (or less) and simple instruction on diet alterations are a very effective way to eliminate or delay T2DM complications. It would be very informative if the authors could include/discuss what a very simple and costeffective plan could do to reduce morbidity and mortality. This is something that the Mexican government and many governments around the world would appreciate (ie. if we spend 20 million dollars on education programs, we reduce or delay this many dialysis visits (and thereby save money!), and prevent this many deaths).

Answer:  We agree with the reviewer on the consideration of economic savings of simple interventions. In the discussion section we have included the next paragraph, as a review of the preventive alternatives to improve the clinical management of diabetic patients:

“On the same vein, for the preventive actions, it could be important to consider the economic aspects of these interventions, given that in Mexico the attention has been concentrated in the pharmacological therapy and not in preventive measures.”

In the next article, we will consider the economic impact of these interventions in the diabetic patients in Mexico; focused on preventive measures.

Reviewer 3 Report

This is a well-written paper that focuses a major health concern, chronic kidney disease in type 2 diabetic patients in a vulnerable population in Mexico.  

While this is a good manuscript, there were minor concerns that the authors need to address.

Page 4 line 152; the definition of chronic kidney disease.  An alternative to albuminuria to determine severity of chronic kidney disease is estimating glomerular filtration rates (eGFR).  GFR was mentioned in line 260 in the Discussion to identify and intervene in patients with early stage chronic kidney disease.  The authors need to explain how their system of denoting the various levels of chronic kidney disease using urinary albumin relates to eGFR values, in Section 2.3 of the Methods.

Some minor English grammar and usage errors were noted: line 87 “So, this patients,…” and line 242 “…the leading cause of decease,…” for example.

Author Response

Reviewer 3.

  • Page 4 line 152; the definition of chronic kidney disease. An alternative to albuminuria to determine severity of chronic kidney disease is estimating glomerular filtration rates (eGFR). GFR was mentioned in line 260 in the Discussion to identify and intervene in patients with early-stage chronic kidney disease. The authors need to explain how their system of denoting the various levels of chronic kidney disease using urinary albumin relates to eGFR values, in Section 2.3 of the Methods.

Answer: Thank you, we added this explanation on the discussion section, instead of methodology section:

“In agreement with the definition of KDIGO International guidelines, and after the adaptation to the clinical practice in the Health system of Mexico, it is suggested albuminuria evaluations in patients with diabetes to estimate cardiovascular risk and its complications; on the other hand, it is suggested that the GFR could be used as a complementary measure to evaluate the existence of hyperfiltration in the diabetic patients. According to Molitch et al. [46] Albuminuria reflects glomerulopathy along with measures of glomerular filtration. People with diabetes may develop only albuminuria, only decreased glomerular filtration, or both. Independent of albuminuria and diabetes, measures of glomerular filtration predict CKD.  Both measures independently increase the risk of mortality “[46].”

  • Some minor English grammar and usage errors were noted: line 87 “So, this patients,…” and line 242 “…the leading cause of decease,…” for example.

Answer: The document was reviewed. All changes were made along the body of the document.
